# PFMNet: Few-Shot Segmentation with Query Feature Enhancement and Multi-Scale Feature Matching

**Jingyao Li** [1], **Lianglun Cheng** [2], **Zewen Zheng** [2,†], **Jiahong Chen** [2,†], **Genping Zhao** [2] and **Zeng Lu** [3,*]

1    School of Automation, Guangdong University of Technology, Guangzhou 510006, China; jingyaoli2055@gmail.com
2    School of Computers, Guangdong University of Technology, Guangzhou 510006, China; llcheng@gdut.edu.cn (L.C.); 2112105305@mail2.gdut.edu.cn (Z.Z.); 2112105306@mail2.gdut.edu.cn (J.C.); genping.zhao@gdut.edu.cn (G.Z.)
3    Dark Matter AI, Xiangjiang International Sci & Tech Center, Guangzhou 510060, China
\*    Correspondence: luzeng@dm-ai.com; Tel.: +86-178-1858-9689
†    These authors contributed equally to this work.

**Abstract:** The datasets in the latest semantic segmentation model often need to be manually labeled for each pixel, which is time-consuming and requires much effort. General models are unable to make better predictions, for new categories of information that have never been seen before, than the few-shot segmentation that has emerged. However, the few-shot segmentation is still faced up with two challenges. One is the inadequate exploration of semantic information conveyed in the high-level features, and the other is the inconsistency of segmenting objects at different scales. To solve these two problems, we have proposed a prior feature matching network (PFMNet). It includes two novel modules: (1) the Query Feature Enhancement Module (QFEM), which makes full use of the high-level semantic information in the support set to enhance the query feature, and (2) the multi-scale feature matching module (MSFMM), which increases the matching probability of multi-scales of objects. Our method achieves an intersection over union average score of 61.3% for one-shot segmentation and 63.4% for five-shot segmentation, which surpasses the state-of-the-art results by 0.5% and 1.5%, respectively.

**Keywords:** few-shot segmentation; semantic segmentation; feature enhancement; multi-scale feature





## 1. Introduction

Few-shot learning has made significant advances in numerous visual comprehension tasks, such as image classification [1–5] and object detection [6–8]. The few-shot learning methods are applied not only to the above aspects but also to the semantic segmentation [9–20] task. However, semantic segmentation methods such as fully convolutional networks (FCN) [15], U-shape convolutional network (U-net) [16], SegNet [9], pyramid scene parsing network (PSPnet) [19], and AMP [13] are not designed to deal with rare and unseen classes. The more annotation with category, the more precise expressions that could have been extracted by the model from the massive data distribution. With a decrease in the number of labeled data, the expression would have shown a progressive reduction. The ultimate goal of few-shot segmentation is to deal with rare and unseen classes from the few-shot images.

Feature processing based on few-shot learning methods can be accomplished by generating weights for the classifier [21], cosine similarity calculation [22–24], or convolutions [25–27] to generate the final prediction. Following that, the existing few-shot segmentation methods [1,21–23,25,28,29] are used as a pre-prior mask to match the test area of the target picture, which learns from the global similarity between the support and query sets. These methods are divided into two categories to process few-shot segmentation tasks. On the one hand, approaches like PANet [23] and PRNet [30] use a metric learning

method to calculate the distance between the query feature and the support prototypes of each category while discriminating between different classes. On the other hand, Adaptive Prototype Learning [22], PMM [31] and PPN [24] show a single prototype is limited by the number of samples, which does not represent the category precisely. At the same time, the prototype obtained by simply taking the mean value is too crude. Therefore, it is recommended to use multiple OSLSM [32] for the learning of a one-shot segmentation model, and the first two-branch network for few-shot segmentation. The main outcome of co-FCN [23] is that it only does some sparse labeling on the original image. The SG-One [33] network contains a backbone network Stem (representing the first three blocks of VGG-16) and two branches. PGNet [34] adds a graph attention mechanism to CANet [25]. However, the feature of originally similar categories will be confused, which reduces the efficiency of image segmentation. In summary, it is typical that classical methods are performed on the model and fix a number of layers to update it into a new model, whereas the other methods use metric learning to get a distribution of large spatial clusters for a few samples. There are still some classical persistent problems in the field of few-shot segmentation.

There are some common issues with existing segmentation methods [21,23,25,28], including inadequate application of semantic information of high-level features, and the other is the inconsistency of segmenting objects' scales. This paper focuses primarily on these two issues.

### 1.1. Inadequate Application of High-Level Features

In few-shot segmentation, since the model uses the support feature to guide the query feature to predict, the feature information of the support feature is essential. However, networks such as CANet [25] simply concatenate the support and query features together. These methods do not specifically enhance the features of the segmented objects in the query image. The support feature and the query feature have the same class of segmentation target information, so the high-level feature of target feature support can be used to enhance the query feature. This makes the model more capable of realizing the segmentation target of the query image. Therefore, we propose a query feature-enhancement module in order to make full use of high-level semantic information of the support set.

### 1.2. Inconsistency of Segmenting Object Scales

Restricted by the quantity of annotations, the problem of inconsistent scale is particularly prominent in few-shot learning. We consider this problem to mainly be caused by inconsistent scale distribution. As shown in Figure 1, the lack of support sample size leads to a sparse scale space (yellow bars), which may be totally divergent from the original distribution (blue bars) of abundant training data. Many existing approaches use multi-scale feature processing to enrich the scale space. For example, PFENet [28] proposes a feature enrichment module (FEM) to solve the problem of space inconsistency caused by global average pooling so that query and support features can exchange information at multiple scales. However, this method does not make any changes to the object scale in the feature map but simply uses a multi-scale approach to enhance the semantic space, which will lead to the problem of inconsistency in segmenting object scales still existing. As shown in Figure 2, compared to the good performance of the medium-scale support feature, the large scale gap leads to the degradation of model performance. Therefore, we propose a multi-scale feature matching module to normalize the segmenting objects' scale to the medium scale and increase the probability of segmenting objects' scale matching in multi-scale feature maps. The details are described in Section 3.3.

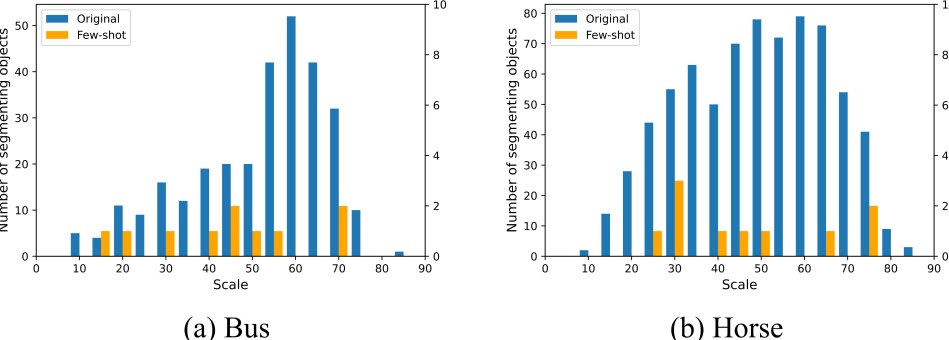

(a) Bus           (b) Horse

**Figure 1.** Proportional distribution maps of image targets for two specific classes: (**a**) bus and (**b**) horse, in PASCAL-5$^i$ (Original) and a 10-shot subset (Few-shot). The object scale is calculated by $S_{obj} = \sqrt{h_{obj}^2 + w_{obj}^2}$, where $h_{obj}$ and $w_{obj}$ are the height and width of the object. From the figure, we can find that the target scale distribution in few-shot learning will deviate from the original datasets.

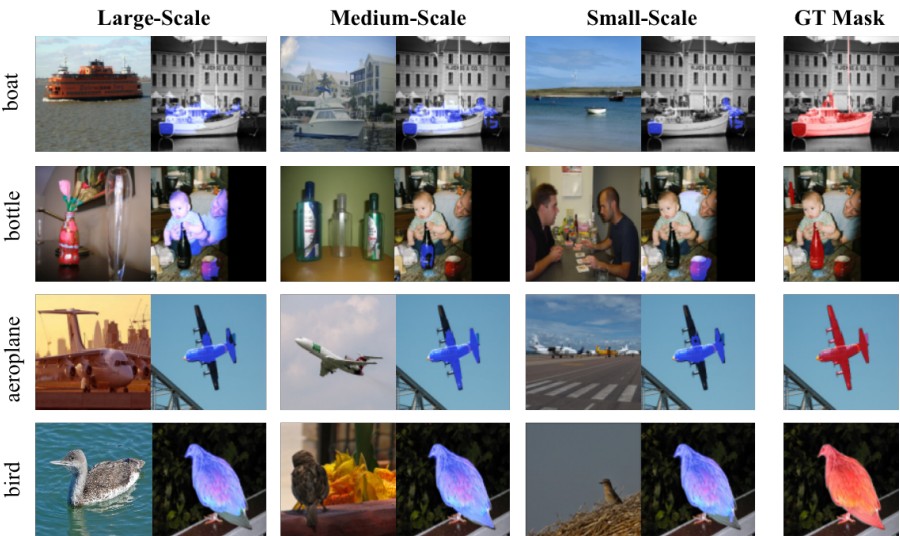

**Figure 2.** The effect of different support target scales on model prediction. Compared to the poor performance of larger target scale gaps, pairs of images with consistent scales are able to make better predictions. For each pair, the left side is the support image, and the right side is the prediction result.

In response to the above problems, our framework adopts the design of two branches to solve the above issues of few-shot segmentation. Based on the proposed query feature enhancement module and multi-scale feature matching module, we propose a new network: prior feature matching network (PFMNet). Our few-shot segmentation model is a generic one. We performed completed experiments on the PASCAL VOC 2012 dataset [21] to validate the effectiveness of our network.

The contributions of this work are as follows.

1.  In order to make use of pixel-level search and dynamic pooling methods, we propose a prior feature matching network (PFMNet) for few-shot segmentation. It is well-learned by the model from fewer annotations.
2.  To apply higher-level features as much as possible, we propose a Query Feature Enhancement Module in order to take full advantage of the high-level semantic information of the support set.
3.  We adopt a more reasonable feature-matching method, a multi-scale feature matching module to normalize the segmenting objects' scale to medium-scale and increase the probability of segmenting objects scales matching in multi-scale feature maps.
4.  Finally, we compare PFMNet with other state-of-the-art methods such as CANet [25], PGNet [34], CRNet [35], SimPropNet [36], LTM [37], RPMM [31], PPNet [24], PFENet [28],

etc. On PASCAL-5$^i$ datasets, our method's performance is better than the state-of-the-art methods under the same conditions.

## 2. Related Work

### 2.1. Semantic Segmentation

Some machine-learning and deep-learning approaches based on semantic knowledge and using technologies from the semantic web are effective in detecting rare classes of elements. Furthermore, semantic segmentation refers to labeling each point of the target class in the image according to "semantics" so that kinds of things can be distinguished in the image. FCN [15] is the pioneer of semantic segmentation, with two main features: the replacement of fully-connected layers with convolutional layers and the fusion of information at different scales. U-net [16] is used to solve simple problems with small samples, such as segmentation of medical films. SegNet [9] and U-net [16] are actually similar in structure, but the difference is that SegNet has not directly fused the information about layers at different scales and pooled with coordinates (index) to solve the problem of information loss until now.

However, semantic segmentation works well on large-sample classes. They are not designed to deal with rare and unseen classes. Compared to models trained with full supervision, our model learns well from weaker annotations.

### 2.2. Application of High-Level Feature

Enhancement of high-level semantic features is an important step in few-shot semantic segmentation. The enhanced high-level features can better guide the query image to segmentation. As mentioned in CANet [25], using the high-level features directly in the network without processing will reduce the performance of the network. Therefore, the high-level semantic features should be enhanced so that the features are nonspecific to a certain class and have better generalizability. Although general data augmentation methods can improve the generalizability of the model, these methods utilize high-level features insufficiently. In previous work, PANet [23] and PFENET [28] obtained an a priori mask after using cosine similarity to measure the distance between the support and query features. The generated a prior mask not only has great generalizability but also assisted the model to better identify targets in query images.

However, the above-mentioned data enhancement of high-level semantic features does not better enhance the similar features of support and query feature. In addition, they do not make full use of the features of support set and do not better solve the problem that high-level features are more specific to a class. Therefore, we propose the query feature enhancement module. Through this module, high-level semantic features are utilized more fully and the model is guided to segment the target of the query more effectively.

### 2.3. Multi-Scale Feature Processing

Since multi-scale feature processing is vital for contextual information understanding, PSPNet [19] utilizes a Pyramid Pooling Module (PPM) for context information aggregation over different regions, which also alleviates the problem of scale inconsistency. DeepLab designs atrous spatial pyramid pooling (ASPP) [38] with filters in different dilution rates to solve the problem of segmenting objects at multiple scales. The model of few-shot segmentation is also introduced. PFENet [28] proposed a Feature Enrichment Module (FEM) that overcomes scale inconsistency by adaptively enriching query features with support features and prior masks.

All the above approaches used different multi-scale feature processing to solve the problem of scale inconsistency. However, these methods have not considered the problems of scale space deviation from the original distribution in few-shot learning and use global average pooling to improve the segmentation result, which will still lead to scale inconsistency and loss of spatial feature information. Moreover, none of these methods consider fluctuations in the target scale of the support set, which makes it difficult for the model to

extract semantic information from different scales. Therefore, on the basis of retaining the FEM of PFENet [28], we propose the Multi-Scale Feature Matching Module (MFMM) to normalize the segmenting objects' scale to medium scale and increase the probability of segmenting objects' scale matching.

## 3. Our Method

In this section, we first briefly describe the details of our proposed prior feature-matching network (PFMNet) in Section 3.1. Then, we present the Query Feature Enhancement Module and the Multi-Scale Feature Matching Module in Sections 3.2 and 3.3, respectively. Finally, we select the cross-entropy loss as our loss function in Section 3.4.

Our goal is to obtain a segmentation model that can quickly learn to perform segmentation from only a few annotated images with new images from the same classes. The train–test split in few-shot learning is in terms of object categories. Thus, all testing categories are unseen during training. In both training and testing, the input is divided into two sets, i.e., the query set $Q$ and the support set $S$. Given $K$ samples from the support set $S$, the goal is to segment the area of unseen class $C_{test}$ from each query image $I_Q$ in the query set.

### 3.1. Prior Feature Matching Network

In order to solve the problems mentioned in Section 2, we propose a prior feature-matching network (PFMNet) that makes full use of high-level semantic information and normalizes the segmenting object scale to medium scale. Figure 3 shows the overall architecture of our network. Specifically, we built a weight-shared framework (imageNet [6] pre-trained ResNet-50) that consists of multiple branches, where one branch is for the query set and the others are for the support set.

After feature extraction, the query-feature-enhance module (QFRM) obtains the similarity matrix with the high-level features and uses the matrix to enhance the query features. We then concatenate the enhanced query feature and similarity matrix as outputs. The multi-scale feature matching module (MFMM) uses a multi-scale structure to process the segmenting object scale in the support feature and increase the probability of matching. We then use a global average pooling to obtain support feature vectors.

After query feature enhancement and multi-scale feature matching, similar to PFENet [28], we use the feature enrichment module (FEM) and a convolution block followed by a classification head to yield the final prediction.

### 3.2. Query Feature Enhancement Module

In order to make full use of the features of support images, instead of using only the mask generated by the similarity matrix of query and image, we use the high-level features to generate a high-level prior feature that guides the model to learn the query features. This feature has sufficient common information about the query feature and the support feature so that the high-level prior feature is beneficial for the model learning it. The process of this module is shown in Figure 4, and there are three steps in total. The first step is to calculate the similarity matrix using cosine similarity, the second step is to recombine the support feature based on the position information of the query features, and the third step is to concatenate it with the enhanced query feature. In Figure 5, we show the effect of the high-level prior feature generated by the module, indicating that the high-level prior feature possesses some information about the position of the query target.

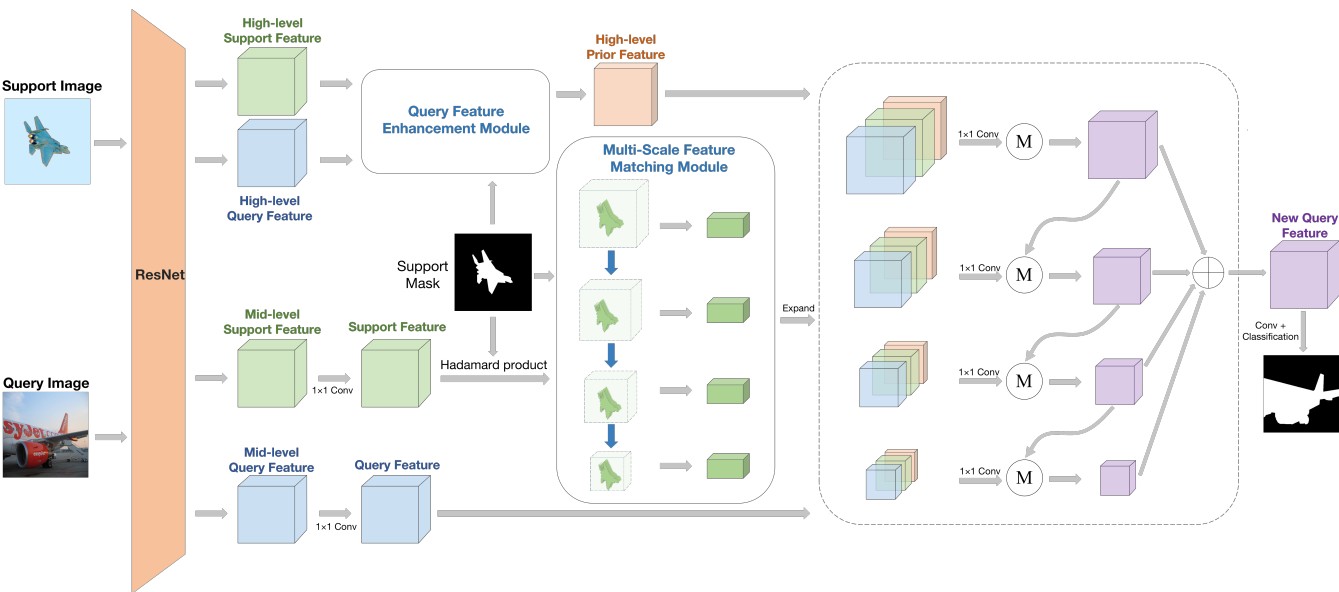

**Figure 3.** Overview of our prior feature-matching network with the query feature enhancement module and multi-scale feature matching module. Blue arrows, circled M, and circled + represent the feature matching, inter-scale merging, and concatenation, respectively. The activation functions are ReLU. PFMNet takes an episode, i.e., the support-query image pair, as input. Based on the mid- and high-level backbone features, the query feature enhance module (QFEM) and the multi-scale feature matching module (MFMM) perform feature enhancement and feature matching on the feature pairs, respectively. Finally, a concatenation, the feature enrichment module (FEM), and a convolution block followed by a classification head are used to yield the final prediction.

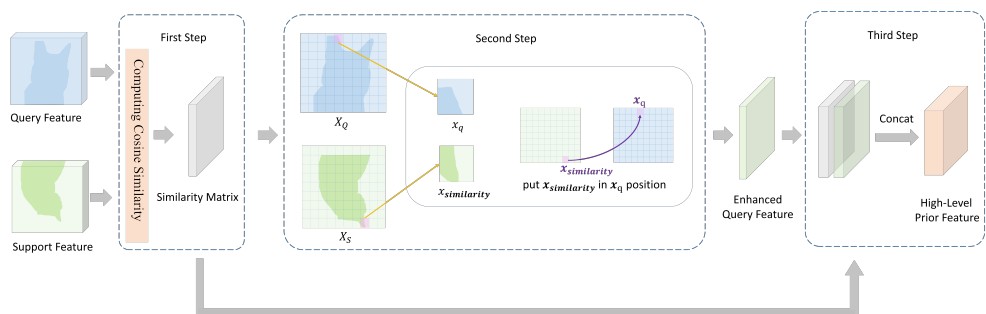

**Figure 4.** Visual illustration step of query feature enhancement module.

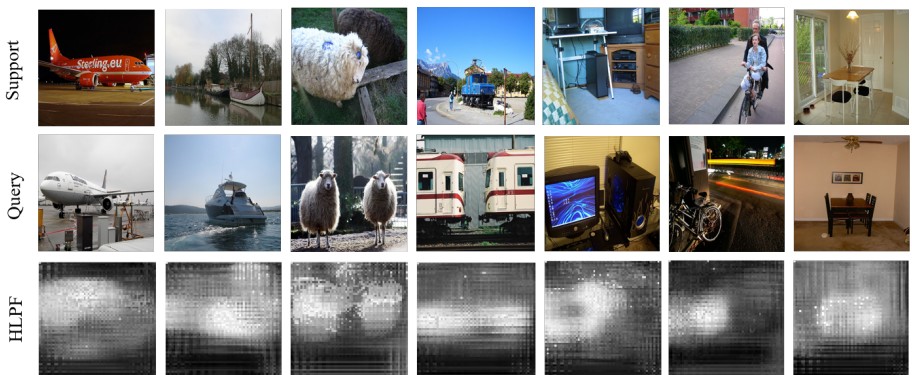

**Figure 5.** Visualization of the QFEM output. This output is a High-Level Prior Feature and is noted as HLPF.

In the first step, through the backbone network, query and support images are extracted for the high-level semantic features $X_Q$ and $X_S$. The sizes of $X_Q$ and $X_S$ are both $\mathbb{R}^{h \times w \times c}$, as shown in Equation (1):

$$X_Q = F(I_Q), X_S = F(I_S) \odot M_S \tag{1}$$

where $I_Q$ and $I_S$ represent the input of support and query images, $F$ denotes the backbone network, $M_S$ denotes the binary support mask, and $\odot$ is the hadamard product.

The cosine similarity is calculated to obtain the similarity matrix $Y$, and the $\cos(x_q, x_s)$ means calculating the cosine similarity between the pixel $x_q \in X_Q$ and $x_s \in X_S$, as shown in Equation (2):

$$\cos(x_q, x_s) = \frac{x_q^T x_s}{\|x_q\| \|x_s\|} \quad q, s \in \{1, 2, 3 \ldots, hw\} \tag{2}$$

where $x_q$ represents a feature vector in $X_Q$, $x_s$ represents a feature vector in $X_S$, $x_q^T$ represents the $x_q$ feature vector transpose, and $\|x_q\|$ and $\|x_s\|$ are the two norms for computing $x_q$ and $x_s$, respectively.

In the second step, the similarity matrix is used to find the similarity that a pixel point in $X_Q$ has to each pixel point in $X_S$. For each $x_q \in X_Q$, we calculate the similarities of this point to each pixel of $X_S$ and get the index of maximum similarity denoted as $c_s$, as shown in Equation (3):

$$c_s = \underset{s \in \{1,2,3\ldots,hw\}}{\arg \max} \left( \cos(x_q, x_s) \right) \tag{3}$$

where $\cos(x_q, x_s)$ is the calculation of the similarity between the $x_q$ and $x_s$ features.

Next, we find a pixel in the $X_Q$ feature denoted as $x_{similarity}$ according to the $c_s$ index, and this pixel is most similar to the $x_q$ pixel. Finally, the $x_{similarity}$ pixels are placed at the position of $x_q$. Each pixel of $X_Q$ completes three steps. and the generated new features are defined as an enhanced query feature. Since the enhanced query feature is generated based on the pixel of the support feature and the position information on the query feature, it has similar features in the target area of query and support. Therefore, it better guides models to learn the information of the segmented target object.

In the third step, the similarity matrix is retained with reference to PFENet [28] because it measures the distance between $X_Q$ and $X_S$. This is a pixel of $X_Q$ has a high value at the position corresponding to similarity matrix $Y$, which means that the pixel is highly correlated with a pixel of the support feature, and this pixel is likely to be in the target region of the query image. Therefore, the similarity matrix $Y$ is normalized and concatenated with the enhanced query feature in the channel dimension to obtain the high-level prior feature as the output of the module. The similarity matrix $Y$ is normalized as shown in Equation (4), so the value of $Y_{normalized}$ is between 0 and 1.

$$Y_{normalized} = \frac{Y - \min(Y)}{\max(Y) - \min(Y)} \tag{4}$$

### 3.3. Multi-Scale Feature Matching Module

In order to solve the problem of inconsistency of segmenting object scales between support and query set, we propose the multi-scale feature matching module (MFMM) to dynamically scale the support feature target area to a medium-scale size according to the scale factor $F$ (as shown in Equation (5)), which allows the model to focus on the semantic information of the segmenting objects and improve the probability of matching in multi-scale feature maps. After average pooling and feature scaling, we take the center of the support mask as the center point to perform regional cropping or padding and use global average pooling on the results of MFMM to generate multiple support feature vectors.

As shown in Figure 6 and Algorithm 1, We first take the support mask as input to calculate the center point and scale factor $F$. Then, we use $2 \times 2$ average pooling on each

feature pyramid size (i.e., [60, 30, 15, 8]) to reduce the feature dimension to ensure that the spatial features of the support feature target area are preserved. The formula is as follows:

$$F = 1.0 + \theta + \frac{S_m \times \theta \times 2}{S_f} \quad \theta \in \{0.1, 0.2 \ldots, 0.4\} \tag{5}$$

where $\theta$ is the scaling control parameter, the function of the above formula is to normalize the scale factor $F$ to the range of $[1 - \theta, 1 + \theta]$, $S_m$ is the target size of the mask, and $S_f$ is the feature size.

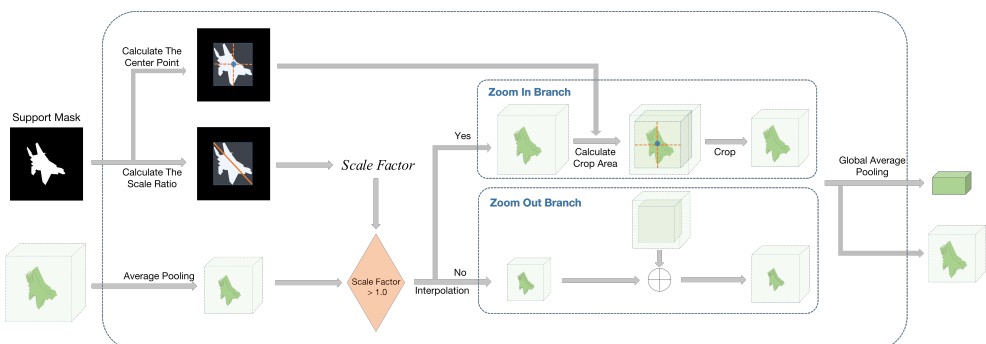

**Figure 6.** A visual illustration of multi-scale feature matching module, which includes two branches: zoom-in branch and zoom-out branch. circled + is a pixel-wise addition. The role of MFMM is to normalize the segmenting object scale according to Scale Factor $F$.

---

**Algorithm 1** Multi-Scale Feature Matching Module

---

**Require:** support feature $F_s$, feature pyramid sizes $PS$, scaling control parameter $\theta$
1: Compute the scale factor $F$ as in Equation (4)
2: **for** every $ps \in PS$ **do**
3:     Let $F_{ps}$ be the support feature after $2 \times 2$ average pooling and interpolation according to previous $F_{ps}$ (first loop is $F_s$) and $F$
4:     **if** $F > 1.0$ **then**
5:         Let $FD_{ps}$ be the $F_{ps}$ cropped according to $TargetCenterPoint$
6:     **else**
7:         Let $FD_{ps}$ be the $F_{ps}$ after paddling
8:     **end if**
9:     Let $FV_{ps}$ be the support feature vector of $FD_{ps}$ after global average pooling
10: **end for**
11: **return** $FV$

---

Then we perform bilinear interpolation scaling and feature processing according to the scale factor $F$. The feature processing process can be divided into two branches: (1) zoom-in branch—calculate the cropping area in the center of the feature target area so as to relatively enlarge the feature target ratio; (2) zoom-out branch: Pad the processed support features to the target size. The purpose of this is to reduce the proportion of the feature target in the support feature. It is worth mentioning that after multi-scale feature matching is completed, we use global average pooling to obtain support feature vector $FV_{ps}$, where $FV_{ps} \in \mathbb{R}^{1 \times 256 \times 1 \times 1}$.

### 3.4. Loss Function

We select the cross entropy loss as our loss function. As shown in Figure 7, the final prediction of our network generates the first loss $\mathcal{L}_1$. The final prediction is generated by concatenating $n = 4$ prediction results of different spatial sizes and then the prediction results generated by the FEM. These n prediction results generate n losses $\mathcal{L}_2^i$. The total loss $\mathcal{L}$ is the weighted sum of $\mathcal{L}_2^i$ and $\mathcal{L}_1$ as:

$$\mathcal{L} = \mathcal{L}_1 + \frac{1}{n}\sum_{i=1}^{n}\mathcal{L}_2^i \tag{6}$$

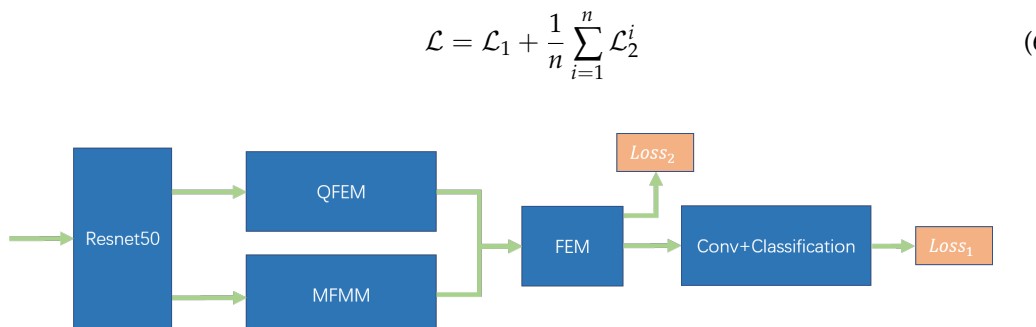

**Figure 7.** The losses come from two aspects of the model.

## 4. Experiments

This sections describes how we validated the effectiveness of our method by experiments. First, we describe the datasets in detail and introduce the experimental environments in hardware and software in Section 4.1. Then, we discuss the implementation details of the experiment in Section 4.2. In Section 4.3, the ablation experiments are explained to explore the optimal combination of our method. Finally, we conducted comparative experiments with the state-of-the-art methods, and the experimental results will be presented and analyzed in Section 4.4.

### 4.1. Description of Datasets and Environments

The datasets on which we conducted our experiments are the PASCAL-$5^i$ datasets, which consists of the PASCAL VOC 2012 and extended annotations from SBD datasets. The 20 classes in PASCAL-$5^i$ [21] are evenly divided into four splits, each containing five classes, as displayed in Table 1. Additionally, we illustrate the percentage of each class in the datasets and show an example image of each class. Models were trained on three splits and evaluated on the other one in a cross-validation fashion. We follow the previous method of sampling 2000 query-support pairs in each evaluation round of training, and we sampled 5000 pairs in testing.

**Table 1.** Testing classes for a four-fold cross-validation test. The training classes of PASCAL-$5^i$, $i = 0$, 1, 2, 3 are disjoint with the testing classes.

| Datasets | Test Classes |
|---|---|
| PASCAL-$5^0$ | aeroplane, bicycle, bird, boat, bottle |
| PASCAL-$5^1$ | bus, car, cat, chair, cow |
| PASCAL-$5^2$ | dining table, dog, horse, motorbike, person |
| PASCAL-$5^3$ | potted plant, sheep, train, tv/monitor |

All of the experiments were conducted on Intel i9-10900XCPU@3.70HGz and GTX 2080Ti GPU, whose graphics memory is 11016MiB. The code of the experiments is implemented by Python $\geq$ 3.6, Numpy $\geq$ 1.17.3 for data preprocessing. The adopted deep-learning framework is Pytorch $\geq$ 1.3.1 and Torchvision $\geq$ 0.4.2. In Figure 8, there are example images and their corresponding segmentation in PASCAL-$5^i$. Bars at the bottom indicate the percentage of the number of different categories in PASCAL-$5^i$, the percentage of which sum to 1.

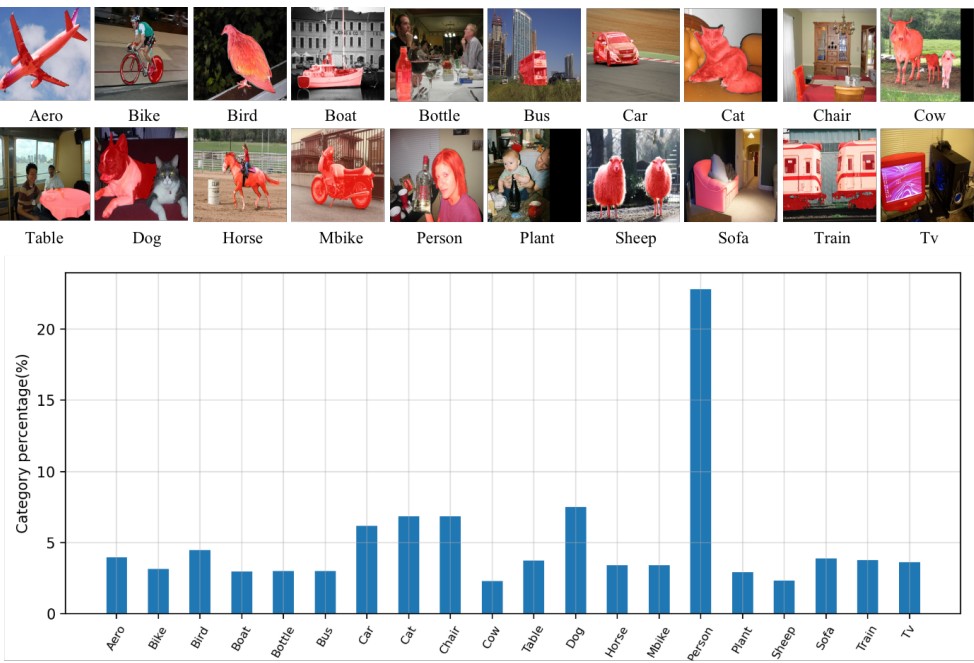

**Figure 8.** Example images and their corresponding segmentation in PASCAL-5$^i$. Bars at the bottom indicate the percentage of the number of different categories in PASCAL-5$^i$.

### 4.2. Implementation Details

In this paper, we select ResNet-50 as our backbone for a fair comparison with other methods. The backbone is initialized with imageNet [6] pretrained weights, and the other layers are initialized with the default setting of PyTorch. Input images are resized to (417, 417) and augmented using random rotation from −10 to 10 degrees. Our model trains are by SGD with a momentum of 0.9. The learning rate is initialized to 0.0025, and weight decay is set to 0.0001.

### 4.3. Evaluation Metrics

We adopted class mean intersection over union (mIoU) as the evaluation metric of our model, which is more indicative of the performance of the model than (the foreground–background IoU) FB-IoU. mIoU is calculated as $\frac{1}{C}\sum_{i=1}^{C} IoU_i$, where $C$ is the number of categories for each fold, which we set to $C = 5$, and $IoU_i$ is the intersection-over-union ratio of the $i$-th category. We average the results of all folds as the final mIoU. In order to be able to compare with other methods, we also give the FB-IoU results. For FB-IoU calculation on each fold, only foreground and background are considered ($C = 2$). However, we do not use FB-IoU as a primary metric for the following reasons, as mentioned by CANet [25]: Firstly, the foreground IoU is more often used in binary segmentation literature. Secondly, as most objects are small relative to the whole image, even though the model fails to segment any objects, the background IoU can still be very high, thus failing to reflect the capability of the model. Thirdly, the numbers of test samples in different classes are not balanced. Ignoring the image categories may lead to a biased result towards the class with more images. Furthermore, we can observe the effectiveness of our model in different classes with the mIoU evaluation metric in Table 2.

**Table 2.** Per-class mIoU results on PASCAL-5$^i$. The models are based on ResNet-50.

| Shot | Aero | Bike | Bird | Boat | Bottle | Bus | Car | Cat | Chair | Cow |
|------|------|------|------|------|--------|-----|-----|-----|-------|-----|
| 1-Shot | 36.9 | 82.2 | 57.3 | 57.1 | 80.8 | 60.9 | 87.2 | 21.2 | 91.2 | 87.9 |
| 5-Shot | 37.5 | 83.1 | 60.5 | 60.5 | 83.9 | 63.2 | 88.7 | 24.9 | 91.2 | 89.1 |

| Table | Dog | Horse | Mbike | Person | Plant | Sheep | Sofa | Train | Tv | MIoU |
|-------|-----|-------|-------|--------|-------|-------|------|-------|-----|------|
| 84.0 | 87.4 | 82.3 | 8.3 | 19.8 | 90.9 | 54.7 | 79.8 | 30.8 | 24.2 | 61.2 |
| 86.2 | 89.3 | 78.5 | 12.9 | 20.4 | 90.4 | 52.7 | 76.0 | 46.0 | 33.0 | 63.4 |

### 4.4. Ablation Experiments

This section describes ablation experiments conducted on two components of the proposed PFMNet, which are the query feature enhancement Modulm (QFEM) and the multi-scale feature matching module (MFMM). The purpose is to demonstrate that the enhanced query feature generated by the high-level semantic features is beneficial for model segmentation. Moreover, multi-scale matching is effective in matching the scales of the segmentation targets of query and support. Note that, to validate the effectiveness of our module and to fit the FEM, we also chose a four-layer structure to incorporate the scale consistency. The baseline in this paper has already demonstrated that using the four-layer branching structure (i.e., [60, 30, 15, 8]) achieves better results. This paper has shown that the four-layer structure enables models to learn query image features in a fine-to-coarse way. The experimental setup follows the description in Section 4.2. All experiments are conducted in the datasets PASCAL-5$^i$ and the fold-0 case. To ensure the highest mIoU of the experimental metric, the mIou is recorded after the loss is sufficiently stabilized.

In this experiment, we validated the effectiveness of the proposed QFEM and MFMM. As displayed in Table 3, the first row shows the mIou of the baseline method. In the second row, the MFMM is applied to the baseline method. In the third row, the QFEM is added to the baseline method. Finally, two modules and the baseline are combined together in the last row. It is worth mentioning that mIoU is one-shot. On the one hand, both "baseline+QFEM" and "MFMM+baseline" are slightly behind baseline. On the other hand, "baseline+MFMM+QFEM" is 1.2% higher than baseline. This means that the effects of the two modules MFMM and QFEM are superimposed on each other, making the total effect greater than the effect of one module alone, because the QFEM outputs a high-level prior feature that fully integrates the similar features from support and query features, and MFMM makes the scale size of the segmentation target easier to recognize by the network. Therefore, the effect of these two modules is enlarged when the outputs of both are put together in the FEM, so that the two modules are used together to achieve a higher mIou than either one alone. In the five-shot, the performance of "baseline+MFMM" and "baseline+QFEM" are both higher than baseline, which shows that MFMM and QFEM are effective for model performance improvement. To sum up, the above experimental results show that MFMM and QFEM both play an important role in model performance.

**Table 3.** MFMM and QFRM. The backbone is ResNet-50 MFMM: Multi-Scale Feature Matching Module. QFRM: Query Feature Enhancement Module.

| Methods | 1-Shot | 5-Shot |
|---------|--------|--------|
| Baseline | 61.7 | 63.1 |
| Baseline+MFMM | 61.0 | 63.2 |
| Baseline+QFEM | 61.3 | 65.1 |
| Baseline+MFMM+QFEM | **62.9** | **65.1** |

### 4.5. Comparative Experiments

In order to assess the utility of PFMNet [28], we compare it with the state-of-the-art few-shot semantic segmentation methods (CANet [25], PGNet [34], CRNet [35], SimProp-

Net [36], LTM [37], RPMM [31], PPNet [24], and PFENet [28]) from recent years. In Table 4, we show the performance of our method in the different cases of one-shot and five-shot. The "-" represents that the corresponding metric score is not given in the paper. For the fairness of the experiment, the models are based on the resnet-50 backbone network and the datasets are based on PASCAL-$5^i$. Note that among all these methods, our proposed network performs better than all present comparison methods in mIoU. Our method's mIoU not only reaches 61.3% in the one-shot case but also up to 63.4% in the five-shot case. This indicates that our method segments image targets more effectively in few-shot semantic segmentation fields. We compare our proposed PFMNet with PFENet [28]. In terms of mIoU, our method can be 0.5% higher than PFENet in the one-shot case and 1.5% higher than PFENet in the five-shot case. Although PFEnet [28] makes use of high-level semantic features, it does not more adequately engage the query to learn support information and does not better enhance the similar features from the support and query features. In addition, it proposes a Feature Enrichment Module to solve the problem of scale inconsistency caused by global average pooling, but the problem of scale inconsistent will still exist in the subsequent feature processing. The QFEM and MFMM in our method solve these problems very well, so the mIoU of our method is higher than it at both one-shot and five-shot.

**Table 4.** Four-fold class mIoU results on ResNet-50 of PASCAL-$5^i$.

| Methods | 1-Shot | | | | |
| --- | --- | --- | --- | --- | --- |
| | Fold-0 | Fold-1 | Fold-2 | Fold-3 | Mean |
| CANet [25] | 52.5 | 65.9 | 51.3 | 51.9 | 55.4 |
| PGNet [34] | 56.0 | 66.9 | 50.6 | 50.4 | 56.0 |
| CRNet [35] | - | - | - | - | 55.7 |
| SimPropNet [36] | 54.9 | 67.3 | 54.5 | 52.0 | 57.2 |
| LTM [37] | 52.8 | 69.6 | 53.2 | 52.3 | 57.0 |
| RPMM [31] | 55.2 | 66.9 | 52.6 | 50.7 | 56.3 |
| PPNet [24] | 47.8 | 58.8 | 53.8 | 45.6 | 51.5 |
| PFENet [28] | 61.7 | 69.5 | 55.4 | **56.3** | 60.8 |
| Ours | **62.9** | **69.7** | **56.4** | 56.1 | **61.3** |
| Methods | 5-Shot | | | | |
| | Fold-0 | Fold-1 | Fold-2 | Fold-3 | Mean |
| CANet [25] | 55.5 | 67.8 | 51.9 | 53.2 | 57.1 |
| PGNet [34] | 54.9 | 67.4 | 51.8 | 53.0 | 56.8 |
| CRNet [35] | - | - | - | - | 58.5 |
| SimPropNet [36] | 57.2 | 68.5 | 58.4 | 56.1 | 60.0 |
| LTM [37] | 57.9 | 69.9 | 56.9 | 57.5 | 60.6 |
| RPMM [31] | 56.3 | 67.3 | 54.5 | 51.0 | 57.3 |
| PPNet [24] | 58.4 | 67.8 | **64.9** | 56.7 | 62.0 |
| PFENet[28] | 63.1 | 70.7 | 55.8 | 57.9 | 61.9 |
| Ours | **65.1** | **71.4** | 57.5 | **59.5** | **63.4** |

We further analyze the results of our comparative experiments by considering each of the folds. Due to the inconsistent data distribution, our results are higher than existing models on most folds and lag just a bit on some folds. In the five-shot case, our model is greater than other methods except for the fold-2, and in the one-shot case, only fold-2 lags a little behind PFENet [28]. This is due to the fact that in the fi e-shot case, the number of support images is higher than in the one-shot case, and the MFMM can better match the support segmentation target to the query target. This makes our method perform better in most cases. In Figure 9, we present the segmentation effects for the same image in the one-shot and five-shot cases. This figure suggests that in the five-shot case, our model pays more attention to the details of the segmentation, and our method is more complete in segmenting the target. We also noticed a significant difference in scores for

different fold cases. In Table 2, we display the scores for each category in the 1-shot and 5-shot cases. This indicates that the segmentation targets for the images in each category are variable, so that the scores for each category are significantly different. In Table 5, we have also provided results for FB-IoU so that our method can be compared with other networks. However, this FB-IoU score only provides some reference and is not a useful evaluation of our model. In Figure 10, we show the images of support, query, ground truth, and our model segmentation results for different categories. This figure presents the segmentation effect of the query. Through our MFMM, we can make the segmentation targets of query and support match as closely as possible. Furthermore, the QFEM provides more support and query information about the segmented target for the network, resulting in full learning of the position information of the query segmented target. Therefore, our method is able to capture the segmentation target of the query image very well, and the segmentation result is very close to ground truth.

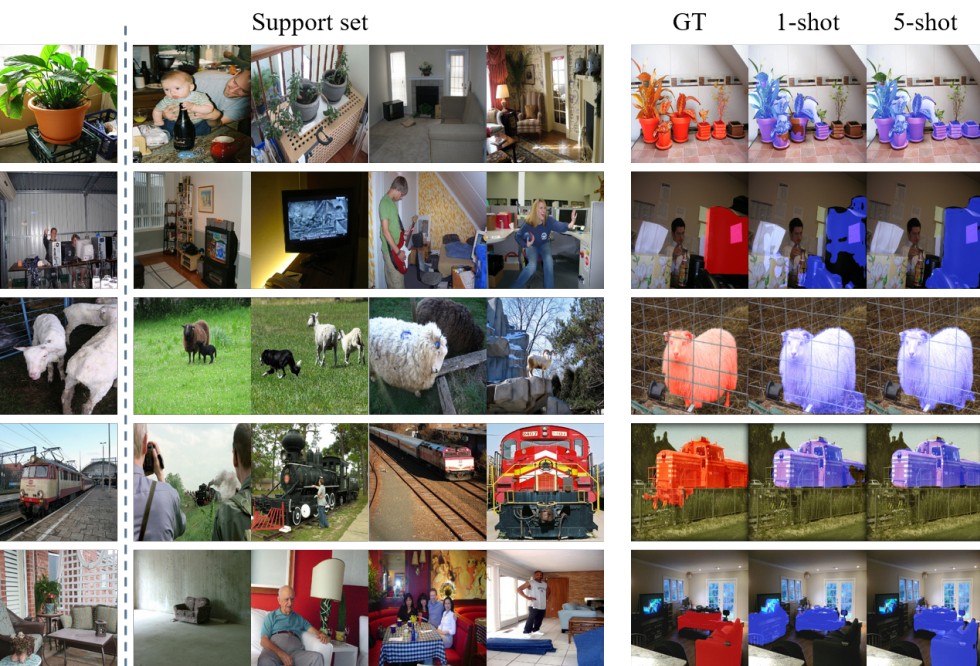

**Figure 9.** PFMNet qualitative results of one-shot and five-shot. The leftmost support of each row is used to generate one-shot results.

**Table 5.** Average FB-IoU results on ResNet-50 of PASCAL-$5^i$.

| Methods | 1-Shot | 5-Shot |
|:---:|:---:|:---:|
| CANet [25] | 66.2 | 69.6 |
| PGNet [34] | 69.9 | 70.5 |
| PFENet [28] | 73.3 | 73.9 |
| Ours | 71.8 | 73.9 |

In order to verify that our model can indeed solve the problem of inconsistent scales, we compared the proposed PFMNet and the baseline with multiple prediction results under the same support image, as shown in Figure 11. By incorporating the proposed MFMM and QFEM, our model surpasses the baseline, reaching a better segmentation effect when the support and query image target scales are quite different. This also proves that by using multi-scale feature matching, our model can better adapt to the problems of inconsistent scale.

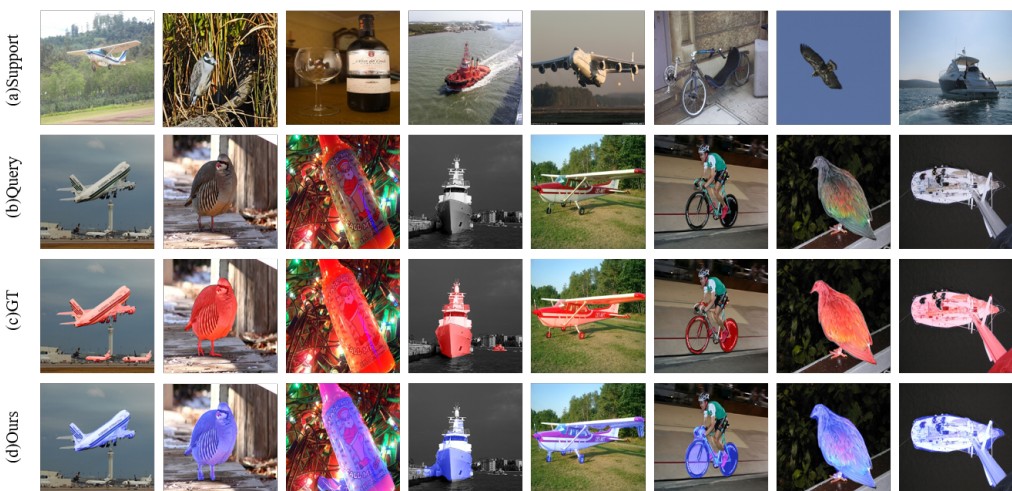

**Figure 10.** Qualitative results of the proposed PFMNet. The samples are from PASCAL-5$^i$. From top to bottom: (**a**) support images, (**b**) query images, (**c**) ground truth of query images, and (**d**) predictions of PFMNet.

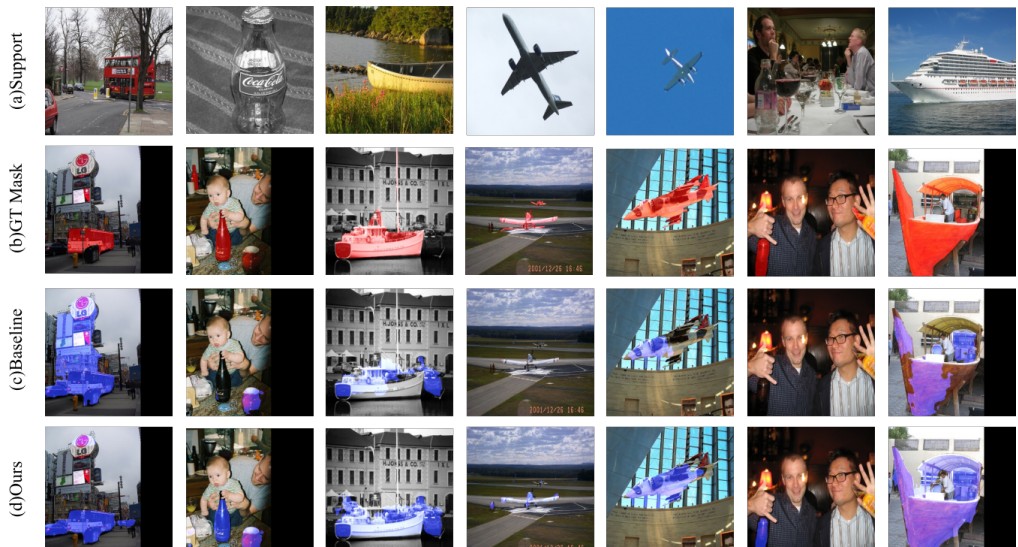

**Figure 11.** Qualitative results of the proposed PFMNet and the baseline under the influence of the large target scale difference between Support and Query images.

## 5. Conclusions and Discussion

In this paper, we propose a prior feature matching network for few-shot segmentation, which consists of the query feature enhancement module (QFEM) and the multi-scale feature matching module (MFMM). The query feature enhancement module uses support for high-level semantic features to enhance the query feature, which boosts the performance of the model. The multi-scale feature matching module solves the inconsistency of segmenting objects scales by leveraging multi-scale normalization technology. With these modules, the experimental results from PASCAL-5$^i$ datasets illustrate that our method achieves significant effects in object scale matching and feature enhancement. In order to assess the utility of PFMNet [28], we compared it with the state-of-the-art few-shot semantic segmentation methods (CANet [25], PGNet [34], CRNet [35], PFENet [28], etc.) in recent years. In Table 4, we show the performance of our method in the different cases of one-shot and five-shot, and our proposed network exceeds all present comparison methods in mIoU. For this result, our method can achieve better performance than other state-of-the-art methods.

Our future work may combine spatial information with existing similar distances to obtain a new high-level feature so as to preserve the local texture (or space) information and adapt to more datasets. In addition, instead of averaging k support features into one feature map, we may introduce a similarity matrix to obtain weights to solve the averaging problem of model features. In addition, we will explore more robust baseline networks and feature fusion methods to perform the improvement of feature processing and modeling.

**Author Contributions:** Methodology, J.L., Z.Z., J.C and Z.L.; software, Z.Z. and J.C.; writing—original draft preparation, J.L., Z.Z. and J.C.; writing—review and editing, L.C. and G.Z. All authors have read and agreed to the published version of the manuscript.

**Funding:** This research was funded by the R&D projects in key areas of Guangdong Province of China grant number 2018B010109007, the R&D projects in key areas of Guangdong Province of China grant number 2019B010153002, the Guangzhou R&D Programme in Key Areas of Science and Technology Projects grant number 202007040006, the Guangdong Jiont Funds of the National Natural Science Foundation of China grant number U1801263, the National High-Resolution Earth Observation Major Project grant number 83-Y40G33-9001-18/20, and the Guangdong Provincial Key Laboratory of Cyber-Physical System(2020B1212060069).

**Data Availability Statement:** Not applicable.

**Conflicts of Interest:** The authors declare no conflict of interest.

## Abbreviations

The following abbreviations are used in this manuscript:

| | |
|---|---|
| PFMNet | prior feature matching network |
| QFEM | query feature enhancement module |
| MFMM | multi-scale feature matching module |
| HLPF | high-level prior feature |
| ResNet | residual network |
| mIoU | mean intersection over union |
| FB-IoU | foreground-background intersection over union |

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
