# Peer review of "PFMNet: Few-Shot Segmentation with Query Feature Enhancement and Multi-Scale Feature Matching"

_information, doi:10.3390/info12100406_

Round 1
Reviewer 1 Report
The paper presents a method to improve few-shot segmentation techniques mainly used to identify rare classes of elements for which there is not enough annotated dataset to perform deep learning efficiently.
The paper is well constructed and focuses on addressing two main problems: the inadequate application of high-level features in few-shot segmentation approaches, and the inconsistency of object segmentation scales.
The introduction and review of related work discuss the main approaches of " few-shot segmentation ", machine learning and deep learning but the approaches based on semantic knowledge and using technologies from the semantic web (such as OWL, SPARQL, SWRL ) are not investigated.
However, these approaches are effective in detecting rare classes of elements because they do not require any training.
Such approaches should be included in the study of related work.
The proposed method is clearly described and the equations are well explained.
Algorithm 1 is not sufficiently explained in the text of the article and is not easily understandable as it stands.
The experiments are adequately detailed, the choices made are explained and correctly justified.
The results show a slight improvement compared to the state-of-the-art, but the article mentions "outperform the state-of-the-art results" line 13 and "achieve better performance than other state-of-the-art methods" line 376. These statements should be more nuanced as the improvement of the results is lite and the experimentations are not carried out on large datasets, which does not allow to draw global conclusions.
The conclusions drawn should be revised and the review of existing work should report on knowledge-based approaches.
Reviewer 2 Report
This paper is focused on few-shot semantic segmentation where authors identify two main challenges with existing art on the problem and devise a new CNN model to address those. Specifically, they propose prior feature matching network (PFMNet) comprised of two novel modules i.e., query feature enhancement module and multi-scale feature matching module to better utilize the feature matching between query and support frames at different scales. The proposed network is demonstrated to outperform existing methods in both 1-shot and 5-shot segmentation settings using PASCAL-5 dataset. A simple ablation is conducted to emphasize the importance of introduced modules. The paper is a good attempt towards bridging the gaps in few-shot segmentation techniques, however, lacks in various aspects to be accepted right away. I suggest a revision for the manuscript and provide my justifications below for this rating.
There are number of English language mistakes. Some of them are identified below with suggested corrections. Other unclear to me are just noted. I have retained corrections only to first two sections. Authors are strongly advised to seek help from a native English speaker to revise their manuscript before submitting it for reviewing.
- Line 6: "and the other is the inconsistency of segmenting objects scales." --> "and the other is the inconsistency of segmenting objects at different scales."
- Line 17: "Few-shot learning have" --> "Few-shot learning has"
- Line 26: "The ultimate goal of few-shot segmentation is to designed to deal with" --> "The ultimate goal of few-shot segmentation is to deal with"
- Line 34: "On the one hand, PANet[24] & PRNet[31]: uses a metric learning method to calculate the distance between the query feature and the support prototypes of each category and meanwhile discriminative for different classes." --> "On the one hand, approaches like PANet [24] and PRNet [31] use a metric learning method to calculate the distance between the query feature and the support prototypes of each category while discriminating between different classes."
- Line 38: "preseccely" --> "precisely"
- Line 46: "it is typically" --> "it is typical"
- Line 58: "The the similar" --> "The similar"
- Line 58: "The similar features of support and query feature are not enhanced." -- confusing sentence.
- Line 60: "so that the model better to reduce the differences between them" -- what better? predict better? -- unclear sentence.
- Line 65: "Due to restricted by the quantity of annotations" --> "Restricted by the quantity of annotations"
- Line 76: "the problem of scale inconsistent will still" --> "the problem of scale inconsistency will still".
- Line 89: "It is well learned by the model from lower annotations." -- unclear sentences, do you mean with fewer annotations?
- Line 94/95: "objects scales" -- unclear, it is "object scales" or "objects' scales" -- the one used by the authors does not make any sense.
- Line 99: "our method achieves outperformed the" --> "our method outperformed the"
- Line 111: "obtain a prior a mask" --> "obtain a prior mask", or do you mean "apriori"
- Line 113: "capability ,but" --> "capability, but"
- Line 130: "have use" --> "have used"
- Line 131: "scale inconsistent" --> "scale inconsistency"
- "Line 131": "have not consider" --> "have not considered"
Please leave space between the work and the references e.g., "image classification[1-5]" should be "image classification [1-5]". Similarly leave the space between terms and their abbreviated versions. For example, "Feature Enrichment Module(FEM)" should be written as "Feature Enrichment Module (FEM)".
Figure 1 is unclear, what does it mean by the diagonal distance? Is it cosine distance, cosine similarity or something else? Please extend the caption of figure 1 to make it independent. A reader do not have to go through the whole section to understand the figure.
Description in lines 74 -- 79 in unclear and is core of their proposed module.
Extend caption of figure 2 to make it independent. It is better to also add the semantic GT class in words instead of just highlighting with red mask i.e., "ship", "bottle", "airplane", and "bird" keywords should be added.
Section 3 should be "Our Method" as authors are introducing only one network, not multiple models.
Clarify figure 3 to identify what does + sign means. The figure can be enhanced further to describe the method in a better manner. More details should also be added to the caption.
Compared to other figures; figure 6 has small scale. Also expand the caption.
It looks like 4 scales are chosen to incorporate the scale consistency, authors should conduct an ablation to study the role of different scales.
Only a single dataset is used for the evaluation, authors should consider other datasets adapted for few-shot segmentation.
Many references in the manuscript are incomplete e.g., [19], [28], [31], [32] etc.
Author Response
Please see the attachment.

This manuscript is a resubmission of an earlier submission. The following is a list of the peer review reports and author responses from that submission.